# Collagen as a Biomaterial for Skin and Corneal Wound Healing

**DOI:** 10.3390/jfb13040249

**Published:** 2022-11-16

**Authors:** Renáta Sklenářová, Naoufal Akla, Meagan Jade Latorre, Jitka Ulrichová, Jana Franková

**Affiliations:** 1Department of Medical Chemistry and Biochemistry, Faculty of Medicine and Dentistry, Palacký University in Olomouc, 775 15 Olomouc, Czech Republic; 2Maisonneuve-Rosemont Hospital Research Centre, Montréal, QC H1T 2M4, Canada; 3Department of Ophthalmology, Université de Montréal, Montréal, QC H3C 3J7, Canada

**Keywords:** extracellular matrix, collagen, biomaterials, wound healing

## Abstract

The cornea and the skin are two organs that form the outer barrier of the human body. When either is injured (e.g., from surgery, physical trauma, or chemical burns), wound healing is initiated to restore integrity. Many cells are activated during wound healing. In particular, fibroblasts that are stimulated often transition into repair fibroblasts or myofibroblasts that synthesize extracellular matrix (ECM) components into the wound area. Control of wound ECM deposition is critical, as a disorganized ECM can block restoration of function. One of the most abundant structural proteins in the mammalian ECM is collagen. Collagen type I is the main component in connective tissues. It can be readily obtained and purified, and short analogs have also been developed for tissue engineering applications, including modulating the wound healing response. This review discusses the effect of several current collagen implants on the stimulation of corneal and skin wound healing. These range from collagen sponges and hydrogels to films and membranes.

## 1. Introduction

The skin and cornea form the outer barrier of the body, providing protection against external influences. Being superficially located, skin and corneal injuries may result from severe infection, burns, physical or chemical trauma, and ultraviolet damage. Although they have similar anatomic structures (e.g., the dermis and stroma are both connective tissues, while the epidermis and corneal epithelium are composed of stratified epithelia [1]), their wound healing mechanisms differ. Severe injuries can result in scarring due mainly to uncontrolled deposition of injury stimulated extracellular matrix (ECM), mainly collagen. To avoid scarring or to allow for scar revisions, a range of natural and artificial polymers based on collagen have been developed to modulate wound healing [2,3,4,5].

Wound healing is an important physiological process, consisting of several phases leading to tissue regeneration after trauma. The wound healing process is complex and depends on the coordinated presence of multiple types of cells, growth factors and cytokines that regulate many cellular processes including growth, migration, differentiation, survival, homeostasis, and morphogenesis [6]. The ECM, discussed in more detail below, also plays an important structural role in this process [7]. 

The most abundant protein present in the human and mammalian body is collagen, and due to its structural role, is also the most widely used protein for tissue engineering of scaffolds [8,9]. Collagens are found in a wide range of organisms [10]. They can be categorized into fibrillar (e.g., collagens I, II, III) and non-fibrillar types (e.g., collagen IV, collagen-like proteins). The fibrillar collagens provides a highly biocompatible and structural environment for cells, tissues and organs with form, stability, and connectivity [11]. Due to its properties, such as its high tensile strength, controllable biodegradability, biocompatibility, availability and the high versatility of its in vitro and in vivo applications [1], it is suitable for the preparation of medical implants such as dressings for burns/wounds, corneal implants, bone filling materials and drug delivery systems. In tissues, collagen is the scaffold material that provides an optimal environment for highly physiologically active cells and cellular components [12].

Currently, biomedical companies manufacture many implants from collagen-rich tissues derived from human or animal sources [13]. These implants differ in structure, crosslinking technology, collagen sources and species and sterilization techniques [14,15,16]. Due to these unique properties, this review discusses the influence of collagen in the cornea and skin in wound healing. Furthermore, its main focus is on collagen-based materials, which are currently being developed, have excellent biocompatible properties and can be further modified. These materials can be used in ophthalmology or in the healing of skin wounds, as a non-immunogenic implant replacing the transplantation of donor cornea or skin tissue.

## 2. Extracellular Matrix

The ECM is a non-cellular structure that surrounds the cells in all tissues. The ECM in mammals comprises approximately 300 [17] proteins that are differentially distributed in individual tissues [18,19,20,21]. The ECM proteins interact with cells and regulate many functions, such as cell proliferation, migration and differentiation [17]. Matrix components bind each other, as well as cell adhesion receptors, forming a complex in all tissues and organs where cells are present. This highly dynamic structural network is mainly composed of proteins (e.g., collagen, elastin, laminin, fibrillin, fibronectin [7,22,23,24,25]) and proteoglycans (e.g., hyaluronate, dermatan sulfate, heparan sulfate, keratan sulfate, and chondroitin sulfate [26]), (Figure 1).

Cell adhesion to the ECM is mainly mediated by integrins. Integrins function as transmembrane receptors and mediate the interaction between the cell cytoskeleton and ECM proteins. [27]. They are capable of interacting with proteins and various signaling molecules [28]. Due to these interactions, integrins can regulate many cellular functions such as cell adhesion, migration, growth and differentiation, and, consequently, can influence the process of tissue repair or regeneration [29,30,31]. Each integrin is composed of two noncovalently-associated transmembrane glycoprotein subunits that are a combination of one to 18 unique α and one to eight β subunits, to form 24 distinct dimers that bind to specific sites of ECM proteins. Integrins αβ heterodimers are divided into four classes (leukocyte, collagen-, Arg-Gly-Asp (RGD)- and laminin-binding integrins) [32]. The α1β1, α2β1, α3β1, α10β1 and α11β1 integrins with characteristic β1 subunits, constitute a subset of the integrin family with an affinity for collagens. Collagen-binding integrins have important functions with respect to the wound healing process: α2β1 affects thrombus formation and α11β1 is a main collagen receptor for collagen remodeling on activated fibroblasts in wounds, fibrotic tissues and the tumor stroma [33]. Therefore, controlling specific cell–ECM interactions offers the possibility to modulate distinct phases of the healing process.

The group of proteins that we will focus on in more detail in this review are collagens. Collagens are the main structural components in various connective tissues and determinants of their tensile strength. Collagens constitute nearly 33% of protein in humans [34]. The structure of collagen consists of three polypeptide chains—α chains—that can form right-handed triple helices, comprising Gly-X-Y repetitions [9]. At positions X and Y, proline and hydroxyproline are often found. The triple helix is stabilized by hydrogen bonds and electrostatic interactions [9,35]. According to their properties, collagens can be divided into several groups: fibril-forming collagens, fibril-associated collagens with interrupted triple helices, network-forming collagens, collagens VI, VII, XXVI and XXVIII, membrane collagens and multi-plexins (collagen XV and XVIII) [8].

Collagens type I and III (fibrillar collagens) are the main structural elements of the dermis followed by fibril-associated collagens type XII, XIV, XVI, and VI [8]. However, the content and distribution of collagens type I and III in the skin varies with age [23]. In the cornea, the major fibril-forming collagens of the ECM are types I and V. The minor collagen types such as collagen III, IV, VII and VIII, are essential in other corneal structures such as basement membranes (BM) and during ECM remodeling [36]. More specifically, collagen type VII anchoring fibrils ensure the adhesion of the epithelium to the underlying structures [37] while disruption of collagen type IV, one of the BM’s principal components, has been shown to lead to several physiological and clinical abnormalities including corneal visual impairments in humans involving corneal opacification [38].

A recent study has shown that collagen type I can bind inflammatory interleukins (IL) such as IL-1β, IL-6, and IL-8 and is capable of forming a physiological wound milieu that supports the healing process [39]. Moreover, collagen type I has exhibited a binding capacity for elastase and matrix metalloproteinase 2 (MMP-2) [40,41].

## 3. The Wound Healing Process

### 3.1. Skin Wound Healing

The skin represents the first barrier protecting the body against injury. Its upper layer, the epidermis, consists mainly of keratinocytes, while the inner, the dermis, consists mainly of fibroblasts [42]. Wound healing is a complex process (Figure 2) involving several successive phases that overlap to some degree: haemostasis, inflammation, proliferation and tissue remodeling [43]. Haemostasis consists of several steps that concludes with the formation of a fibrin clot. 

In vertebrates, almost every cell (with exceptions of avascular tissues such as the cornea) is located within a distance of 100 microns from a capillary. In the highly vascularized skin, the capillaries are damaged during injuries [44,45]. The first step is a localized transient vasoconstriction of blood vessels to limit blood loss, followed by activated thrombocytes binding to exposed collagen and endothelial lining to form a platelet plug. This temporary seal is further solidified by the intrinsic and extrinsic coagulation clotting cascade which forms a more robust fibrin mesh at the wounded site [45,46]. These processes are assisted by multiple sources and chemical mediators including endothelin, serotonin, Von Willebrand factor, Adenosine diphosphate, integrins and collagen-binding glycoprotein VI as well as the release of chemotactic and growth factors such as transforming growth factor β (TGF-β), contributing to the restoration of normal homeostasis after trauma [47,48,49]. 

The inflammatory response and phagocytosis stimulated by the wounding acts to clear pathogens, foreign bodies or damaged tissue present in the wound [49] allowing for repair through proliferation and remodeling that involves interactions between ECM and cells. Once bleeding has stopped, the phase of inflammation is established by the widening of blood vessels. This hyperemic state is initiated by locally released chemicals from plasma, damaged cells, or pre-formed cellular or synthesized pro-inflammatory mediators [46]. These factors act to increase blood flow and permeability to plasma molecules and immune cells within the wound [50]. This is paramount to support anabolic processes necessary to fully perfuse and repair damaged tissues by providing oxygenation and nutrients for new tissue to arise [50,51,52]. Pattern recognition receptors (PRRs) can sense damage- or pathogen-associated molecular patterns (DAMPs and PAMPs) and promote the release of many proinflammatory cytokines [53,54]. The release of inflammatory cytokines and growth factors, such as IL-1, IL-6, IL-8, TNF-α, platelet derived growth factor (PDGF) and endothelial cell growth factor (VEGF), increase the influx of inflammatory cells [50]. Vascular permeability in and around the injured tissue facilitates the infiltration of leukocytes (neutrophils and monocytes) that clean up the wound by phagocytizing cell debris and pathogens in the interstitial space. The presence of elevated cytokines and growth factors support the migration and proliferation of skin cells and the synthesis of ECM molecules necessary for wound regeneration [55]. In the injured skin, matrix metalloproteinase 1 (MMP-1) production is also induced by keratinocytes that bind to type I collagen in the dermis through α2 and β1 integrins [56]. Overproduction of these degradative matrix metalloproteinases (MMPs) could damage host healthy tissue around the wound area [57]. 

After a few days, a transition from the inflammatory phase to the proliferation phase is necessary for tissue formation and efficient wound closure. Tissue remodeling and establishment of new blood vessels through angiogenesis is critical in wound healing and takes place concurrently during all phases of the reparative process [51]. The onset of angiogenesis is upregulated by several factors, mainly VEGF and TGF-β [51]. Another protein that strongly stimulates this process is collagen type I. It appears that the binding of α1β1/α2β1 integrin receptors on the surface of endothelial cells is crucial for its angiogenic activity [58]. Fibroblasts that migrate into the wound contribute to granulation tissue formation. Granulation tissue is composed of new connective tissue and tiny blood vessels that proliferate profusely and produce the matrix proteins [59]. Subsequently, fibroblasts transform into contractile myofibroblast phenotypes that attach to fibronectin and collagen in the extracellular matrix [60]. In the proliferative phase, the key event is the production of TGF-β that affects the transcription genes of collagen, decreases the production of MMPs and increases the levels of protease inhibitors—TIMPs [61]. The reepithelization process, which is important for restoring an intact epidermis, is mainly provided by keratinocytes which migrate along the fibrin blood clot and on to the surface of the granulation tissue [62]. 

The remodeling process is the last phase and may take several months. The tensile strength of the wound is gradually increased when wound healing collagen type III is replaced by ordinary collagen type I. The final result is a fully matured scar [63,64].

### 3.2. Corneal Wound Healing

While both the skin and cornea are ectoderm-derived, the cornea possesses two striking differences, its optical transparency and avascularity [4]. The cornea is also the most innervated and sensitive tissue in the human body [65,66]. It consists of three main cellular layers—an outermost epithelial layer, middle stroma layer and an innermost endothelial layer [67] (Figure 2). These layers are separated by acellular layers, the Bowman’s membrane between the epithelium and stroma and the Dua’s (pre-Decemet’s layer) and Descemet’s membrane (DM) between the stroma and endothelium. Each cellular layer has different regenerative capacities and differ in their involvement in corneal wound healing. There is well-documented cross talk between epithelial and stromal cells during the wound healing process that leads to restoration of corneal integrity [65]. Maintaining corneal transparency and avascularity during wound healing is essential to preserve optimal vision [66,68]. 

The inflammatory response paradigm of the cornea differs from that of the skin as multiple anti-angiogenic molecules normally present in the cornea contribute to its “avascular privilege”. The avascularity also contributes to its “immune privilege”, also referred to as “lymphangiogenic/haemangiogenic privilege” as there are very few mature immune cells within the cornea [69]. It has been reported that the expression of soluble VEGF receptors in the healthy cornea (e.g., sVEGFR-1, -2, -3) [4,70], binds and inhibits the activity of vascular VEGF and prevents the in-growth of blood vessels or lymphatics [71]. Other growth factors that contribute to the avascularity and immune privilege include thrombospondins (TSP-1 and TSP-2) [72] and PEDF [73]. Anatomical components such as the ECM-dense0 collagen fibers and derived bioactive precursor fragments such as endostatin, tumstatin, arresten, canstatin, neostatin, restin and angiostatins also maintain the barrier to vessels and immune cells [71]. Anti-angiogenic factors found in the aqueous humor and the limbal region are also thought to contribute to the avascular nature of this tissue [71,74,75]. 

The sensitive dynamic between pro-angiogenic and anti-angiogenic substances can promote or inhibit corneal haemangiogenesis and lymphangiogenesis. Important orchestraters of this delicate equilibrium are MMPs. More than 16 MMPs have been identified in the cornea which include collagenases (e.g., MMP-1, 8 and 13) gelatinases (e.g., MMP-2 and 9) and membrane type MMPs (e.g., MMP-14) which can act as either pro or anti-angiogenic modulators [76]. In some corneal pathologies (infection, inflammation, ischemia, degeneration, trauma, surgery, herpes) [77], the balance can be shifted towards the pro-angiogenic status if sustained, leading to neovascularization [71,74,78]. In such cases of inflammation and severe pathology, the immune privilege of the cornea is gone [79]. A key molecule involved in pathogenic corneal neovascularization is VEGF [80]. Expression of VEGF has been attributed to stimulation by inflammatory cytokines, notably interleukins-1,6 and tumor necrosis factor-alpha (IL-1, IL-6, TNF-α) [81,82,83]. Interestingly in some pathologies, corneal neovascularization occurs in parallel with epithelial hyperplasia, disruption of the Bowman layer, disorganized stromal collagen fibers and fibroblast activation [4].

#### 3.2.1. Corneal Epithelial Healing

Corneal epithelial wounding can stimulate an acute inflammatory response at the limbus leading to the accumulation of leukocytes, the migration of neutrophils, dendritic cells and macrophages, and the influx of lymphocytes within the stroma and the wounded epithelium [65,84]. 

The entire corneal epithelium can be replenished in approximately seven to ten days [4,85]. This process is accelerated when the injury is superficial such as with most corneal abrasions that heal without any complications [4,86]. Stem cells and, in particular, progenitor cells such as the transient amplifying cells of the limbal epithelium contribute to the rapid healing [82,86,87,88]. Corneal epithelial wound healing involves several key phases: adhesion of the ingrowing cells to the BM, migration to cover the wound, cell proliferation and differentiation [87,89,90]. The epithelium and keratocytes around the wound remain viable after a surface injury and reconstitute a new BM. However, they will undergo apoptosis or necrosis in severe damage or deep wounds, prolonging the wound closure [68,91]. While several major collagen types are differentially distributed across all corneal layers, the BM is composed mainly of collagen type IV which provides adherence to epithelium [92], other types may also be found (IV, VII, XII, XV, XVII, XVIII) and play an important role in the morphology and pathology of corneal disease [37]. The collagen of the underlying Bowman layer, consisting mainly of collagen types I, followed by III, V and XII, does not regenerate after injury by keratocytes, but can be substituted by a structurally different Bowman’s-like layer [92,93].

#### 3.2.2. Corneal Stromal Healing

In continuum with the Bowman’s layer is the corneal stroma. It is the thickest layer of the cornea that is made mostly of collagen fibrils (types I and V) [92,94]. Epithelial and stromal cells interact through the BM. The first stromal response after epithelial or epi-stromal injury is mediated by IL-1 and TNF-α release and is characterized by keratocyte apoptosis in the area of the injury. This is followed by the activation of adjacent quiescent stromal keratocytes and their conversion into fibroblasts [68,95]. In addition to apoptosis, in some cases, such as in severe alkali burns, necrosis can occur [96]. The release of IL-1, TNF-α, IL-6, CXC chemokines, and MCP-1 by the injured cells promotes the infiltration of inflammatory cells into the wound area [4,68,97]. Within 24 hours, activated fibroblasts migrate to the wound, subsequently proliferate and then trans-differentiate into contractile opaque myofibroblasts, the prominent repair phenotype [95] mediating wound closure and contraction [4]. To facilitate tissue repair and wound closure, the differentiated myofibroblasts proliferate and migrate towards the site of injury and deposit excessive amounts of ECM collagen, including type I, III, IV, and V and matrix stabilizing proteoglycan [36]. Fibroblasts and myofibroblasts deposit several components of the ECM including collagen type III, fibronectin, tenascin, elastin and proteoglycans which enables the migration of fibroblasts [4,68,98]. Although still debated [99], collagen III is weakly expressed in the normal cornea, but its expression increases several-fold during wound healing. It forms fibers by association with collagen type I and therefore serves as one of the main markers of stromal matrix remodeling observed after corneal injury [100]. However, as large amounts of collagen III are not normally present in stroma, its elaboration after stromal injury may interfere with collagen fibril lamellar assembly during remodeling and contribute to scar formation [68,101]. During corneal wound healing, the abnormal deposition of the various types of collagens at the wrong place or the wrong time including types III, IV and XII are responsible for corneal opacities and scarring [68,101].

Multiple factors influence either a regenerative or fibrotic healing outcome. These include multiple secreted growth factors, particularly the TGFβ’s from the epithelium, tears and to a lesser extent the stromal cells. Normal levels of these regulators are reestablished with BM regeneration during physiological corneal repair after wound closure, after which myofibroblasts undergo apoptosis. Persistence of wound healing growth factors and myofibroblasts can otherwise result in uncontrolled disorganized matrix secretion in which there is excess collagen deposition. The resulting disorganized matrix becomes resistant to collagenase remodeling, leading to scarring and haze [68,95,102,103,104]. Myofibroblast persistence also hinders the regenerative capacity of the cornea including the BM and corneal nerves [68]. Corneal fibrosis is considered irreversible with few preventive options and may require surgery or transplantation, if human donor corneas are available [103]. Corneal stromal regeneration is very slow and can take months or years. The normal form and function of the stroma are slowly restored by the removal of infiltrating inflammatory cells, resorption of abnormal scar ECM, and, finally, repopulation of stroma by keratocytes and elaboration of ECM components found in the healthy cornea [68]. 

A comparable posterior stromal wound healing response can occur after a sustained injury to the endothelium and its associated Descemet’s membrane (DM), but how myofibroblasts form within the stroma after endothelial injury is unclear [4,68,95].

#### 3.2.3. Corneal Endothelial Healing

The human corneal endothelium separates the aqueous humor from the rest of the corneal layers [105]. It consists of a flattened monolayer resting on a basal membrane called Descemet’s membrane, which itself lies next to the recently discovered Dua’s layer [106]. Relative to the BM, DM has reduced regenerative capacity and is mainly composed of non-fibrillar collagens type VIII and type IV produced by corneal endothelial cells [100,107,108]. However, only collagen IV remains adjacent to endothelial cells in both infant and adults, while collagen VIII becomes displaced towards the stromal side with age [108]. The integrity of the corneal endothelium and DM are critical for the corneal fluid homeostasis that in turn controls corneal transparency. The endothelium is responsible for the bidirectional exchange between the cornea and aqueous humor of fluids, nutrients and bioactive molecules such as growth factors or cytokines [100,105,107,108]. Extensive loss of endothelial cells caused by disease, injury or aging can lead to corneal edema due to excessive fluid accumulation [4]. Corneal endothelial cells have very limited regenerative potential and proliferative capacity, but they can nevertheless expand, migrate and undergo endothelial–mesenchymal transition (EnMT) acquiring a phenotype to repair wounds [4,36,65,100]. This transformation is mediated by factors such as TGF-β, FGF-2, IL-1 and involves NFκB activation and upregulation of collagen type I. In severe pathologies such as alkali burns or infections, EnMT results in aberrant ECM deposition posterior to DM and endothelial fibrosis that leads to sight threatening complications [4,68,108,109,110,111].

**Figure 2 jfb-13-00249-f002:**
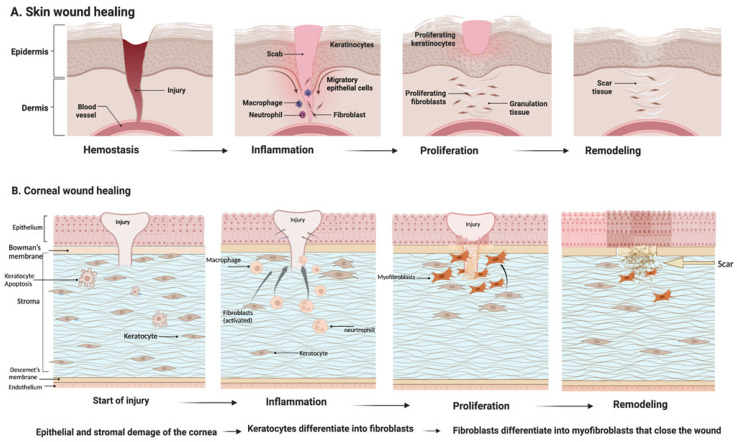
The Wound healing process. Wound healing is an important physiological process to maintain the integrity of the skin or cornea after trauma. Diagram (**A**) shows the course of skin healing—individual phases such as hemostasis, inflammation, proliferation and skin remodeling. Figures created with BioRender.com (accessed on 1 October 2022). (**A**) Modified according to Ref. [112] and (**B**) [95].

## 4. Biomaterials

Because the human body has limited regenerative capabilities, current treatment options to replace damaged tissue and organs is by donor tissue and organ transplantation [113]. Over the last two decades, research efforts in the field of regenerative medicine and tissue engineering have increasingly focused on the replacement of transplantation using bioimplants made of biocompatible materials. Individual biomaterials are used in different ways, depending on their applications and location e.g., hydrogels or scaffolds that are made of 3D bioprinted tissue [108] and are designed based on understanding of the process of healing individual tissues. Basically, these materials must not be toxic and must be biocompatible with the patients’ tissues [114]. Prepared materials need to have mechanical, chemical, or physical properties that are appropriate for those of the target organ, as well the ability to promote seamless integration into the host. Many of these materials contain main proteins of the ECM, such as collagen, fibronectin or ECM´s polysaccharides [115,116].

## 5. Collagen Biomaterials for Wound Healing

Collagen is the most common protein in the human body and is suitable for the production of biomaterials used in tissue engineering [117]. It contributes to the mechanical strength and elasticity of tissues and acts as a natural substrate for cellular attachment, proliferation, and differentiation. It can mediate a lot of pro-regenerative physiological interactions during the complex wound healing process ranging, from angiogenesis to re-epithelialization [117]. In addition, it can bind and inactivate excessive amounts of MMPs that occur in non-healing wounds [118,119]. Excellent biocompatibility and intrinsic biodegradability by endogenous collagenases make exogenous collagen ideal biomaterial for use in biomedical applications [120]. Collagen-based biomaterials can be classified into two categories—i) decellularized collagen matrices that retain the original tissue properties and ECM structure, and ii) scaffolds prepared via extraction, purification, and collagen polymerization [1,121]. All prepared biomaterials must meet the limits of assessment of biocompatibility and toxicity of the material, in vitro and subsequently in vivo.

The final biological application of collagen products depends on the type of the injury and biomaterial character. Frequently used forms of biomaterial (Figure 3) include gels, membranes, sponges, hollow fiber tubing and spheres [120]. Examples of commercial collagen products for skin healing are presented in Table 1.

Collagen for use as biomaterials in wound healing and regenerative medicine are also within the research realm and are being developed and tested in various forms for use in skin and corneal wound healing. There are numerous variations, so only a selection will be discussed.

Currently, nano forms of collagen are also being explored, produced by electrostatic spinning, which has advantages over the conventional three-dimensional (3D) design of collagen, mainly due to its nanoscale, which contributes to a higher surface-to-volume ratio and helps to withstand large loads with minimal stress [132].

The ideal materials for cornea healing should replicate the properties of the natural cornea and show seamless biointegration with host tissues, be resistant to infection, have excellent optical properties [133]. The commercial biomaterials-based implants that allow corneal vision restoration after limited wound healing include the keratoprostheses (KPros) or artificial corneas. KPros that are currently used are the Boston type 1 keratoprosthesis (Boston KPro-1) for eyes with good ocular surface and the Osteo-odonto-keratoprosthesis (OOKP) and Boston KPro-2 for treatment of dry eyes and damaged ocular surfaces based on polymethylmethacrylate (PMMA) [133,134]. However, because of their very limited capacity for regeneration and the prevalence of severe side effects, these are only used in end-stage eyes. Presently, research has focused primarily on the production of collagen-based or inspired corneal implants that promote regeneration. The most common collagens are type I and III in the form of hydrogels, which have already been evaluated in clinical trials. In vitro tests of collagen sponges are also in progress [134,135,136].

### 5.1. Collagen Sponges

Commercial collagen sponges are insoluble, prepared mostly by lyophilized aqueous collagen solutions. Their use reduces the formation of scarring and promotes hemostasis, so they are used as a biomaterial for skin wound healing [137,138,139]. These sponges are capable of absorbing large amounts of tissue exudate, and will adhere to a wet wound, maintain a moist environment and protect wounds from mechanical trauma and bacterial infection [140]. Implanted collagen sponges are infiltrated with tissue containing glycosaminoglycans (GAGs), fibronectin, and new collagen, followed by various cells (e.g., sponge implantation in burn wounds leads to a rapid recovery of the skin due to an intense infiltration of neutrophils in the sponge) [141]. These sponges are also effective depots for the storing and releasing of exogenous growth factors (such as TGF [142]) to wounds and are also suitable for the short-term delivery of antibiotics. They are especially useful in wound healing because their wet-strength allows for them to be sutured to soft tissue and provides a template for new tissue growth. Immunohistochemistry results published by Chang at al. showed that collagen type I and III expression was increased in the wounds treated with collagen-containing sponges [143]. Regeneration of the epidermis and collagen fiber deposition was observed also in a study by Cheng at al., where they used a collagen sponge with carboxymethyl chitosan on burn wounds [144].

Collagen sponges have also been investigated as substrates for the culturing of human corneal cells and potential stromal replacements. Keratocytes cultured on collagen type I sponges showed increased ECM synthesis and cell proliferation, which could improve the healing process [145,146].

### 5.2. Hydrogels

Hydrogels are aggregates of hydrophilic polymers that can absorb or contain large amounts of fluids, even ≥90% of water [147,148]. Collagen hydrogels are often considered the most promising wound healing candidate owing to their mechanical properties, gelling ability, stability, biocompatibility and low toxicity [122]. These hydrogels have a wide range of uses, they present a large uniform surface area for supporting cell growth and their high water content allows for the exchange of gases, nutrients and waste products and helps them serve as depots for drugs. Sturdier hydrogels can also act as barriers against bacterial contamination [149]. Another advantage of the hydrogel over conventional wound dressings is their reported capacity for reducing pain through a cooling effect, while injectable hydrogels have low adhesion to the tissue and do not cause pain when removed [129]. Collagen hydrogels often incorporate other biological matrix molecules such as glycoasminoglycans (GAGs). Based on the type of GAGs used (hyaluronic acid (HA), heparin, heparan sulfate (HS), chondroitin sulfate (CS), dermatan sulfate (DS), and keratan sulfate (KS)), the hydrogels have different functionalities for in vivo use [121]. Collagen-hyaluronan-based hydrogels have also been used to develop in vitro organotypic models to mimic healthy or malignant extracellular matrices [150]. 

Injectable hydrogels, which have good fluidity and consist of collagen I and hyaluronic acid (COL-HA), have been proposed for non-healing wounds [151]. Collagen has been modified with a number of other substances to ensure that the bioimplants have good mechanical properties. Fabrication of three-dimensional (3D) composite scaffolds based on collagen and chitosan in different proportions have been reported [152]. Chitosan–collagen hydrogels have good haemostatic (blood coagulation) capacity and make promising wound dressing [153,154]. These hybrid hydrogels possess good thermal stability, injectability, and pH sensitivity [155] and have superior mechanical strength compared to chitosan-only hydrogels [156]. 

Plastic compressed type I collagen hydrogels with incorporated keratinocytes and fibroblasts are being tested in a phase two clinical trial as denovoSkinTM [157,158]. These constructs are characterized by an epidermis that is properly stratified during transplantation, appear to develop a functional basement membrane and dermo-epidermal junction, and exhibit an almost normal functional dermis [157].

Collagen hydrogels are also being examined for use in ophthalmology [159,160]. Collagen type I, the primary molecule of the native cornea, is a suitable biomaterial for corneal tissue engineering applications. These hydrogels are frequently used as models for studying neovascularization in the cornea. The importance of collagen I in angiogenesis is evident because collagen degradation releases proangiogenic factors [161,162]. Furthermore, the type of collagen used has an impact on the physical properties of the final product. For instance, the addition of soluble tropocollagen improves transparency and strength. While collagen (type I and type III) hydrogels have comparable tensile strength and elasticity, collagen type III hydrogels tend to be slightly mechanically and optically superior [163]. 

Several in vitro and in vivo studies have reported that short ECM-mimicking peptides can stimulate regeneration in a range of organ systems. As an alternative to full-length collagen, short collagen-like peptides (CLPs) have been used to fabricate soft hydrogels [154] or have been conjugated to synthetic polymer for better mechanical strength as implants [164,165]. 

One synthetic polymer that is commonly used to conjugate CLP is polyethylene glycol (PEG) which is chemically inert [164,166,167]. CLP-PEG implants can be enhanced with 2-methacryloyloxyethyl phosphorylcholine (MPC), an artificial lipid that suppress the inflammation. The resulting CLP-PEG-MPC implants improved the reduction of corneal swelling, haze, and neovascularization in comparison with CLP-PEG implants [136]. PEG–collagen hydrogels also enable the encapsulation of viable mesenchymal stem cells (MSCs) that could be a promising method in cases of ocular inflammatory diseases, such as alkali burn injury [167].

Full-length recombinant human collagen type III (RHCIII), which behaves like native collagen, has been used to fabricate implants on their own, or as composites with polymers of MPC. The resulting RHCIII–MPC hydrogels are robust and have been shown to allow precision femtosecond laser cutting to produce complementing implants and host surgical beds for subsequent tissue welding [168]. Another material suitable for femtosecond laser surgery is a bioengineered porcine collagen (BPC) platform based on high-purity, medical-grade collagen extracted from porcine skin. The BPC is highly compatible with cell ingrowth and has favorable optical and mechanical properties. It can replace a portion of the native corneal stroma with rapid wound healing promoting a transparent cornea [160]. However, xenogeneic materials extracted from animals must be used cautiously as they can cause severe allergies [169], or as seen in the recent COVID-19 pandemic, zoonotic transmission can occur [170].

### 5.3. Other Applications: Films and Membranes

Collagen is one of the most effective materials for the preparation of membranes and films, due to their ease of manufacture. Collagen membranes that are mechanically strong have been widely used in medicine and dentistry, due to their high biocompatibility and resorbability [119]. Recently, biodegradable collagen membranes have been applied in guided bone regeneration with comparable outcomes to non-resorbable membranes [171]. Many collagen films are prepared by a process of crosslinking and some of these are mixed with other polymers that improve their properties. Currently, one of the most studied films for wound healing are collagen–chitosan films that are doubly crosslinked, e.g., using tannic acid and genipin. When compared with other collagen–polymer blends or to pure collagen, they show higher antimicrobial activity and improved physicochemical properties [172]. These membranes and films can be hydrated to allow the exchange of gases (such as O_2_ and CO_2_). Such dual crosslinked polymeric films are being increasingly examined for use in ophthalmology, especially as temporary implants for injured corneas [173]. Moreover, films combining collagen with hyaluronic acid or with chitosan have found applications in cosmetics and tissue engineering [153,174].

An overview of the currently developed material discussed in this review (mentioned in Section 5.1, Section 5.2 and Section 5.3) is also given in Table 2.

## 6. Conclusions

This review highlights several examples of collagen-based or inspired biomaterials that are being used for skin and corneal tissue wound healing and regenerative medicine. These biomaterials are well-tolerated, facilitate rapid wound closure, promote cell proliferation and also the formation of new ECM in a very short time. In addition, collagen-based biomaterials can be modified by bioactive substances to suppress inflammatory responses. They can also be combined with other materials to fine-tune the physical, chemical and mechanical properties of the resulting biomaterials and, hence, their functionality. The collagen-based hydrogels, sponges, films and membranes presented here are some examples that have excellent biocompatible healing properties and can be further modified based on their use.

## Figures and Tables

**Figure 1 jfb-13-00249-f001:**
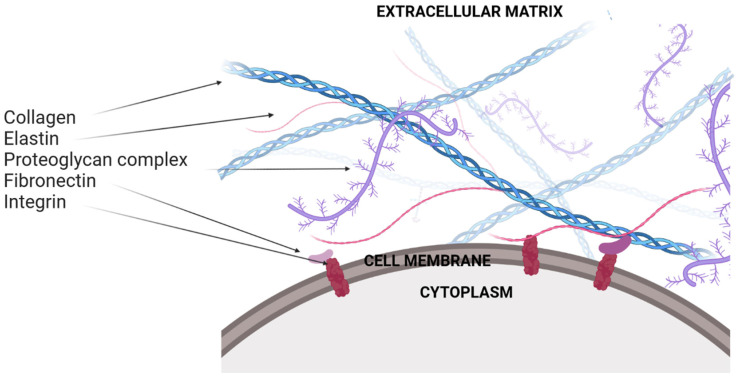
Diagrammatic overview of the extracellular matrix and its major components (e.g., collagens, elastin, proteoglycan complexes, fibronectin, and interacting integrins). Created with BioRender.com (accessed on 1 October 2022).

**Figure 3 jfb-13-00249-f003:**
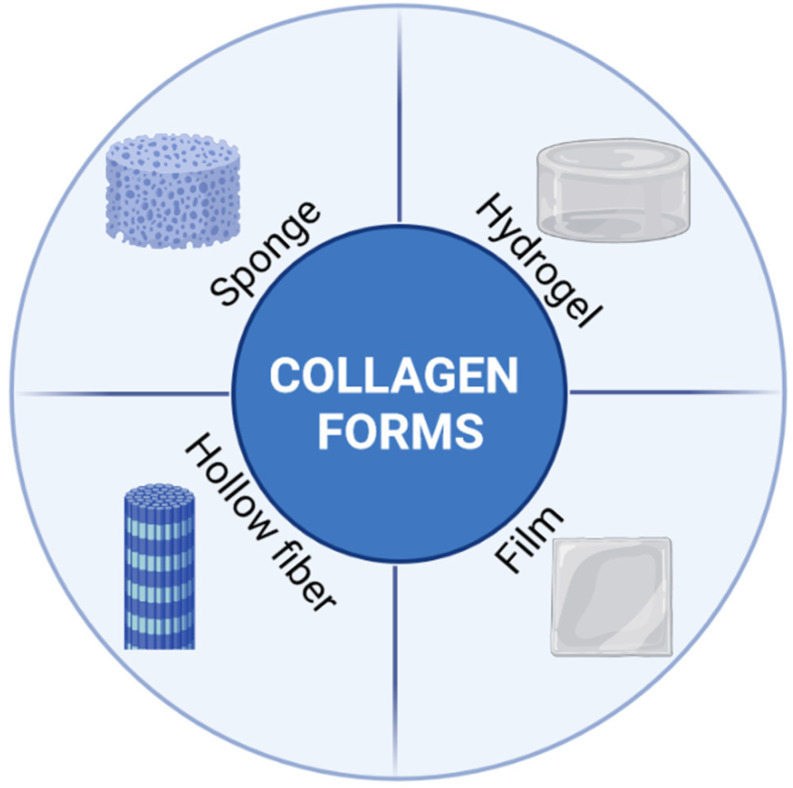
The main collagen forms used in biomaterials. Created with BioRender.com (accessed on 1 October 2022).

**Table 1 jfb-13-00249-t001:** Summary of the overview of commercial collagen materials used for skin healing.

Wound Dressing Materials/Collagen Form	Product	References
Collagen sponges	Avitene™ Ultrafoam™ Collagen Sponge	[122,123]
	GENTA-COLL^®^ resorb	[124]
	Helistat^®^	[122,125]
	Microfibrillar Dressing Instat™ Mch Collagen	[122]
	Collatamp^®^-G	[126,127]
Collagen films and membranes	SkinTemp^®^ II	[126,128]
	Fibracol^®^	[128,129,130]
	Promogran™/Promogran Prisma^®^	[126,128,130,131]
	CollaSorb^®^	[129,130]
	BIOPAD™	[131]
	Puracol^®^ Plus/Puracol^®^ Plus Ag	[128,131]
	ColActive^®^ Plus/ColActive^®^ Plus Ag	[131]
	DermaCol™	[128]
Collagen hydrogels	HYCOL^®^	[128]
	Collatek^®^	[128]
	CellerateRX^®^	[128,129]

**Table 2 jfb-13-00249-t002:** Summarizes the overview of currently developed non-commercial collagen materials discussed in this review.

Wound Dressing Materials/Collagen Form	Non-Commercial Product	References
Collagen sponges	Collagen sponge form jellyfish	[138]
	Recombinant collagen hemostatic sponge	[139]
	Collagen sponge form porcine, bovine and human skin	[140]
	Platelet-rich plasma–collagen sponge	[143]
	Carboxymethyl chitosan–collagen peptides sponge	[144]
	Collagen I sponge	[145,146]
Collagen hydrogels	Collagen–hyaluronan hydrogels	[150,151]
	Collagen–chitosan hydrogels	[152,153,154,155,156]
	Collagen hydrogels with incorporated cells	[157,158]
	Collagen hydrogels (type I and III)	[159,160,161,162,163]
	Collagen-like peptides hydrogels	[136,164,165,166,167]
	Recombinant human collagen type III hydrogels	[168]
	Collagen hydrogels (form porcine)	[169]
Collagen films and membranes	Collagen membranes	[119,171]
	Collagen–chitosan films	[172,173,174]

## Data Availability

Not applicable.

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
