# Peer review of "Collagen as a Biomaterial for Skin and Corneal Wound Healing"

_jfb, 2022, doi:10.3390/jfb13040249_

Round 1

Reviewer 1 Report

Collagen-based or -inspired biomaterials for wound management is always an interesting topic. The manuscript reviews the biomaterials for skin and corneal tissue wound healing and regenerative medicine. More discussion should be added to the manuscript before can be accepted for publication. Please find my comments below:

1.     Line 49. “being highly versatile” in the case of what? Please elaborate in the manuscript.

2.     Line 54—58. Should be combined into one paragraph. Moreover, please extend this paragraph by explaining the novelty of this research as compared with already published such as Mathew-Steiner et al., Bioengineering 2021, 8(5), 63; doi: 10.3390/bioengineering8050063. Also explain why corneal injury is of interest in this review.

3.     Kindly define all abbreviations, especially those biomolecules such as interleukin (IL), etc.

4.     Careful with the representation of ECM in figure 1. Does it mean fibroblasts are also among the ECMs?

5.     Representations of collagen-based material in the form of ‘gels, membranes, sponges, hollow fiber tubing and spheres’ should be presented. Authors my take the pictures from the published literatures as long as they obtain copyright clearance which is easy to do. Otherwise, make the representation by your own.

6.     General outcome (such as wound healing rate, side effects, etc.) along with the study design should be elaborated in Table 1. Current table is not acceptable.

7.     Make another table on non-commercial collagen-based materials. Presenting the material, modification, study design (in vivo, in vitro, clinical trial), general characteristics, and general outcome.

8.     Risks and potential toxicities of collagen-based materials (especially those modified using polymeric blend and crosslinkers) should be discussed.

9.     What about the risk of neovascularization due to the pro-angiogenic property of collagen. 

Author Response

Reviewer 1

Collagen-based or -inspired biomaterials for wound management is always an interesting topic. The manuscript reviews the biomaterials for skin and corneal tissue wound healing and regenerative medicine. More discussion should be added to the manuscript before can be accepted for publication. Please find my comments below:

Thank you for your kind revisions and suggestions for improvement of our paper. We appreciate all your time given to our article.

  1. Line 49. “being highly versatile” in the case of what? Please elaborate in the manuscript.

Thank you for the comment, it was corrected in the text.

  1. Line 54—58. Should be combined into one paragraph. Moreover, please extend this paragraph by explaining the novelty of this research as compared with already published such as Mathew-Steiner et al., Bioengineering 2021, 8(5), 63; doi: 10.3390/bioengineering8050063. Also explain why corneal injury is of interest in this review.

Thank you for the comment, it was corrected in the text.

  1. Kindly define all abbreviations, especially those biomolecules such as interleukin (IL), etc.

Thank you for the comment, it was corrected in the text.

  1. Careful with the representation of ECM in figure 1. Does it mean fibroblasts are also among the ECMs?

Thank you for the comment, Figure 1 has been corrected. Fibroblasts were excluded from the picture.

  1. Representations of collagen-based material in the form of ‘gels, membranes, sponges, hollow fibre tubing and spheres’ should be presented. Authors my take the pictures from the published literatures as long as they obtain copyright clearance which is easy to do. Otherwise, make the representation by your own.

The graphic abstract and Figure 1 were entirely drawn by the first author, Renata Sklenarova. Figure 2 was entirely drawn by author Meagan JL.

  1. General outcome (such as wound healing rate, side effects, etc.) along with the study design should be elaborated in Table 1. Current table is not acceptable.

We thank the reviewer for this comment, but unfortunately, the general outcome for commercial materials cannot be summarized in the table for several reasons.

Clinical studies are not publicly available for almost the material (some of them are the part of the patents). At the same time, these materials are used for different types of injuries (e.g. diabetic wounds, non-healing wounds, decubitus and etc.), and each patient responds to treatment individually. Due to these variations, it is not possible to generally summarize healing time or side effects from the clinical point of view in the table.

  1. Make another table on non-commercial collagen-based materials. Presenting the material, modification, study design (in vivo, in vitro, clinical trial), general characteristics, and general outcome.

There are many non-commercial collagen-based materials that are currently developed or are being in process in scientific research. We suppose that these materials cannot be uniformly summarized in a table and for that reason we focused only on the commercial one.

  1. Risks and potential toxicities of collagen-based materials (especially those modified using polymeric blend and crosslinkers) should be discussed.

Thank you for the comment. Evaluation of the toxicity of all collagen materials (and especially those that are modified using polymer mixtures and cross-linking agents) is determined in vitro (on cells suitable for the final application of the material) and subsequently studied in vivo. Thus, for the final wound healing materials are chosen only non-toxic of them.

  1. What about the risk of neovascularization due to the pro-angiogenic property of collagen.

The importance of collagen I in angiogenesis is evident because after collagen degradation are released proangiogenic factors. In wound healing is the formation of new blood vessels an important part of tissue repair. In the process of corneal healing, neovascularization is undesirable.

In cornea exists a balance between angiogenic factors, such as fibroblast growth factor (FGF) and vascular endothelial growth factor (VEGF), and antiangiogenic molecules, such as angiostatin, endostatin, or pigment epithelium–derived factor (PEDF). Furthermore, many collagen implants enriched with substances (e.g. MPC) that have the ability to inhibit corneal neovascularization are being developed, and are described in the manuscript.

Reviewer 2 Report

The paper Collagen as a biomaterial for skin and corneal wound healing by SklenáÅ™ová et al. describes the use of collagen as a biomaterial for aiding in wound repair. The manuscript is generally well written, easy to read, and provides good background information on the importance of collagen in skin and corneal structures, the wound healing process, and examples of collagen-based biomaterials. While the manuscript provides general information, there is room for improvement.

This manuscript can be published after some minor corrections are made.

The purpose of the review as stated in the Abstract and Introduction sections needs to be reconciled. In the Abstract one is lead to believe that collagen-based biomaterials will be discussed, but in the Introduction section it is stated that the review will discuss on the influence of collagen in wound repair. To some extent, both of these topics are discussed; more of this review focuses on the ECM and wound healing, and less on the biomaterials. These statements should be reconciled to clearly state the intention of this review.

Generally, there needs to be a more in-depth discussion of the results of the primary literature that is cited. While the manuscript is well written, it reads like general description of the role of collagen in biology, wound repair, and a few mentions of some biomaterials. The positive outcome of this approach is that this is a great place to begin reading for non-experts in the field. Given the number of collagen-based products mentioned in Table 2, I would like to see more discussion of these references in the main body of the text.

There is much information provided regarding corneal wound healing processes, yet there are few examples provided in the biomaterials section. If there are in fact only a few examples of collagen biomaterials available for corneal healing applications, I question the whether the volume of information provided on corneal wound healing in Section 3 is needed.

Section 5.3 Films and membranes contain no mention of skin/corneal wound healing. The title of the review suggests that skin/corneal wound healing is the topic of the review, so this section needs to (i) be expanded to include examples of films and membranes in skin/corneal healing applications, (ii) be removed, or (iii) be placed in a section of "other applications"

Figure 2 should be placed closer to the first mention of the figure.

Minor points to consider:

Line 55 - change differing to differ.

Line 110 - Please define the acronym MMP-2. The acronym MMP is used frequently throughout the manuscript without being defined.

Line 203 - Be consistent with acronym styles for closely related names (TSP-1 and TSP2).

Line 346-347 - This sentence is poorly worded and difficult to understand.

Line 404 - Ref 147 does not direct to work by Chang et al. as stated.

Author Response

Reviewer 2

The paper Collagen as a biomaterial for skin and corneal wound healing by SklenáÅ™ová et al. describes the use of collagen as a biomaterial for aiding in wound repair. The manuscript is generally well written, easy to read, and provides good background information on the importance of collagen in skin and corneal structures, the wound healing process, and examples of collagen-based biomaterials. While the manuscript provides general information, there is room for improvement.

This manuscript can be published after some minor corrections are made.

The purpose of the review as stated in the Abstract and Introduction sections needs to be reconciled. In the Abstract one is lead to believe that collagen-based biomaterials will be discussed, but in the Introduction section it is stated that the review will discuss on the influence of collagen in wound repair. To some extent, both of these topics are discussed; more of this review focuses on the ECM and wound healing, and less on the biomaterials. These statements should be reconciled to clearly state the intention of this review.

Generally, there needs to be a more in-depth discussion of the results of the primary literature that is cited. While the manuscript is well written, it reads like general description of the role of collagen in biology, wound repair, and a few mentions of some biomaterials. The positive outcome of this approach is that this is a great place to begin reading for non-experts in the field. Given the number of collagen-based products mentioned in Table 2, I would like to see more discussion of these references in the main body of the text.

There is much information provided regarding corneal wound healing processes, yet there are few examples provided in the biomaterials section. If there are in fact only a few examples of collagen biomaterials available for corneal healing applications, I question the whether the volume of information provided on corneal wound healing in Section 3 is needed.

Section 5.3 Films and membranes contain no mention of skin/corneal wound healing. The title of the review suggests that skin/corneal wound healing is the topic of the review, so this section needs to (i) be expanded to include examples of films and membranes in skin/corneal healing applications, (ii) be removed, or (iii) be placed in a section of "other applications"

Figure 2 should be placed closer to the first mention of the figure.

Thank you for your kind revisions and suggestions for improving our manuscript. We appreciate all your time given to our article. In the Abstract and Introduction sections, we aligned with the purpose of the review.

There are only few materials for corneal healing but their development is currently rising up. We believe that this review will be a good starting point for many authors in the field of wound healing process.

Minor points to consider:

Line 55 - change differing to differ.

Thank you for the comment, it was corrected in the text.

Line 110 - Please define the acronym MMP-2. The acronym MMP is used frequently throughout the manuscript without being defined.

Thank you for the comment, it was corrected in the text.

Line 203 - Be consistent with acronym styles for closely related names (TSP-1 and TSP2).

Thank you for the comment, it was corrected in the text.

Line 346-347 - This sentence is poorly worded and difficult to understand.

Thank you for the note, it was corrected in the text.

Line 404 - Ref 147 does not direct to work by Chang et al. as stated.

We apologize for the mistake and thank the reviewer for this comment. It was corrected in the text.

Round 2

Reviewer 1 Report

I thank authors for their responses. However, I still feel the manuscript could not add significant contribution to science if the followings are not fulfilled:

a. Sample pictures of Collagen-based material in its various forms should be presented. 

b. Make another table on non-commercial collagen-based materials. Presenting the material, modification, study design (in vivo, in vitro, clinical trial), general characteristics, and general outcome.

Authors argue that the non-uniformity does not allow the authors to tabulate the summaries. I cannot accept this reason. Authors of review articles should find strategies for overcoming the stated problem.

Author Response

Thank authors for their responses. However, I still feel the manuscript could not add significant contribution to science if the followings are not fulfilled:

Dear reviewer, thank you very much for spending time to reading our manuscript and making suggestions how to improve that.

  1. Sample pictures of Collagen-based material in its various forms should be presented.

Dear reviewer, thank you very much for this suggestion. We prepared the figure that includes the general forms of collagen materials (figure 3).

  1. Make another table on non-commercial collagen-based materials. Presenting the material, modification, study design (in vivo, in vitro, clinical trial), general characteristics, and general outcome.

Dear reviewer, we include the Table 2 that summarise the overview of non-commercial collagen materials discussed in our review. It is included in the text.

Authors argue that the non-uniformity does not allow the authors to tabulate the summaries. I cannot accept this reason. Authors of review articles should find strategies for overcoming the stated problem.

Round 3

Reviewer 1 Report

The manuscript has been improved into a satisfactory level. It can be accepted for publication.